# Sliding Mode Control with Feedforward Compensation for a Soft Manipulator That Considers Environment Contact Constraints

Yinglong Chen [1,2,*], Qiang Sun [1], Jia Wang [3], Junhao Zhang [1], Pengyu Zhao [1] and Yongjun Gong [1]

[1] Naval Architecture and Ocean Engineering College, Dalian Maritime University, Dalian 116000, China; s2549724961@163.com (Q.S.); dmuzjh@163.com (J.Z.); pengyuzhao@dlmu.edu.cn (P.Z.); yongjungong@163.com (Y.G.)
[2] State Key Laboratory of Fluid Power and Mechatronic Systems, Zhejiang University, Hangzhou 310027, China
[3] JiuJiang 707 SCI&TECH Co., Ltd., Jiujiang 332007, China; 13607022275@163.com
[*] Correspondence: chenyinglong@dlmu.edu.cn

**Abstract:** Soft manipulators have desirable environmental compatibility because of their pliability. However, this pliability also brings challenges to modeling and control when considering contact or collision with the environment. In previous work, we established several mathematical models for describing fluidic soft manipulators under environmental effects and verified their accuracy. However, the controller design for a soft manipulator is still a significant challenge, especially under the conditions of environmental contact. In this paper, we build upon our previously established work by conducting feedforward compensation for a soft manipulator under contact constraints and designing a sliding mode controller based on an operational space dynamics model. Then, we combine the feedforward compensation model with the sliding mode controller to realize accurate position control of the soft manipulator. Finally, simulation and experimental results show that this controller can accurately and effectively control the position of the soft manipulator.

**Keywords:** contact constraint; motion control; sliding mode controller; soft manipulator feedforward compensation

## 1. Introduction

Recently, with the development of 3D printing and new soft materials, soft robots have become a research hotspot [1–3]. Soft robots are characterized by strong deformability. According to the bionic principle, soft robots can be divided into soft worms, soft robotic fish, and soft manipulators [4,5]. Soft manipulators are an important research branch within soft robots. They have infinite degrees of freedom and a flexible structure, which can be used to handle fragile objects and execute tasks in complex environments [6–9]. Therefore, they have broad prospects in industry, surgery, defense, and other fields. However, the pliability of soft manipulators also presents challenges in their modeling and control. At present, the kinematic modeling methods for soft manipulators typically include the constant curvature model [10], piecewise constant curvature model [11], variable curvature model [12], Cosserat rod model [13], and more. Many methods have been developed for the kinematics modeling of soft robots. And the dynamic modeling methods of soft manipulators have also been widely studied. Mahl et al. established a dynamic model of the soft manipulator based on the Euler–Lagrange method and verified the accuracy of the model through simulation and experiments, respectively [14]. Xun et al. proposed an underwater dynamic model of the soft manipulator based on the Kane method. This dynamic model considers the interaction between soft manipulators and complex fluid environments [15]. In addition, the dynamic modeling method based on the Cosserat theory is also widely used [16–18]. There is relatively little research on using the Newton–Euler

iteration method for the dynamic modeling of soft manipulators. In order to expand the dynamics research on the soft manipulator, we proposed a new dynamic modeling method for soft manipulators based on the Newton–Euler iteration method. And we verified the accuracy of this method through experiments [19].

After establishing the kinematics and dynamics models, the next step is to design the controller based on the established model to realize the control task of the soft manipulator. Bailly et al. obtained the required actuators or configuration space variables through the direct inversion of the differential inverse kinematics (IK). Using this model, they developed a simple linear closed-loop controller [20]. Wang et al. designed a differential kinematics controller based on the variable curvature model, which performed visual servo control of cable-driven soft manipulators [21]. On this basis, Wang et al. established a dynamic model of the soft manipulator in a complex underwater environment. Based on the dynamic model, they designed an adaptive controller and verified its control performance through trajectory-tracking experiments [22]. Falkenhahn et al. proposed an integrated controller based on joint space dynamics, which used an inversion method to convert the generalized torque in the dynamic model into actuator pressure. They carried out experiments to verify the controller performance. But it did not take into account the dynamic characteristics of the actuator, so its performance was limited [23]. Ivanescu et al. developed a sliding mode controller and a fuzzy controller for two cooperative super-redundant robots. However, those works were limited to simulation [24]. Santina et al. simplified the soft manipulator model to a rigid link model and established a dynamic model. Based on this model, they designed an impedance controller that can control the end effector of the soft manipulator to move along the contact surface when it comes into contact with the environment. And the control accuracy of this controller was verified through experiments and simulations, respectively [25]. The environmental contact problem is an important research topic within soft manipulators. Marchese et al. proposed a static model of a soft manipulator considering the environmental contact but found that the contact force had to be obtained through sensors [26]. In order to achieve precise control of soft manipulators under environmental interference, Yip et al. proposed a model-free hybrid positioning method [27], which drove the soft manipulator to move freely in an unknown environment [28]. Toscano et al. proposed a feedback control strategy based on a dynamic model of the soft manipulator for realizing environmental contact. This method did not need to measure and estimate the contact force [29]. The contact of the soft manipulator with the environment will change its original movement posture so that it cannot move to the desired position. Although some research has considered the environmental contact problem of soft manipulators, designing a controller that can control the soft manipulator to move to the desired position when disturbed by the environment remains a huge challenge. Therefore, we design a controller that can achieve position control of the soft arm under environmental constraints.

In this paper, we combine feedforward and feedback methods to control a soft manipulator. First, we establish a static feedforward model with contact constraints. In addition, we derive an operational space dynamics model for the soft manipulator. On this basis, we design a sliding mode controller and use the Lyapunov stability criterion to analyze its stability. Finally, simulation and experimental results show that the controller can accurately and effectively execute the position controls of the soft manipulator. The controller designed in this article can be applied to fluidic-driven/pneumatic soft robots, but it cannot be applied to electric-driven or other types of soft robots.

## 2. Static Feedforward Model

### 2.1. Static Feedforward Model

First, we describe the soft manipulator used in this study. The length of the soft manipulator is 0.56 m, and the weight is 2.5 kg. The soft manipulator is composed of two sections. Each section contains three soft units, and a rubber hose passes through the soft manipulator to drive each soft unit. The soft unit matrix is made of ELASTOSIL®M4601-type silica gel. This type of silica gel has a density of 1.01 g/mm$^2$, a tensile strength of

5 N/mm$^2$, an elongation at break of 700%, a tear strength greater than 30 N/mm, and a linear shrinkage rate of less than 1%. When driving the silicone matrix, it undergoes both radial expansion and axial elongation simultaneously. For soft units, constraining their radial deformation can increase their driving efficiency. Therefore, we wrap a layer of elastic fabric outside the soft unit to limit its radial expansion. Next, we will introduce the static feedforward model of the soft manipulator. The feedforward model can be used to compensate for the feedback control and improve its accuracy. However, the kinematic feedforward control is limited by the low precision of the kinematics model, which does not consider the influence of gravity and elastic force; the compensation effect is thus poor. This paper establishes a static feedforward model based on the kinematic feedforward model. This model considers the influence of driving pressure, gravity, and elastic force. In this paper, we define the following two assumptions.

**Assumption 1.** *We use two soft manipulators in a series, and we use a piecewise constant curvature model to divide each soft manipulator into n/2 continuous arcs along the axis direction. We treat each arc approximately as having a constant curvature. Therefore, the entire soft manipulator can be divided into n continuous arcs.*

**Assumption 2.** *The driving pressure acts uniformly on each soft unit, and the direction of action is always along the axis direction.*

### 2.1.1. The Mapping $f_{gx}$ from Joint Space to Operational Space and Its Inverse Mapping $f_{xg}$

As shown in Figure 1, in the local coordinate system where the *i*-th segment of the soft manipulator is located, its shape characteristics are defined by the arc length, bending angle, and deflection angle in the joint space. The arc length $l_i$ represents the arc length of the central axis of the *i*-th segment of the soft manipulator. Bending angle $\theta_i$ is the center angle corresponding to the arc length $l_i$. We define the plane on which the axis of the soft manipulator is located when it is bent as the deflection plane PlaneB. Deflection angle $\varphi_i$ is the angle between the deflection plane PlaneB and the positive half-axis of the *x*-axis in the current local coordinate system.

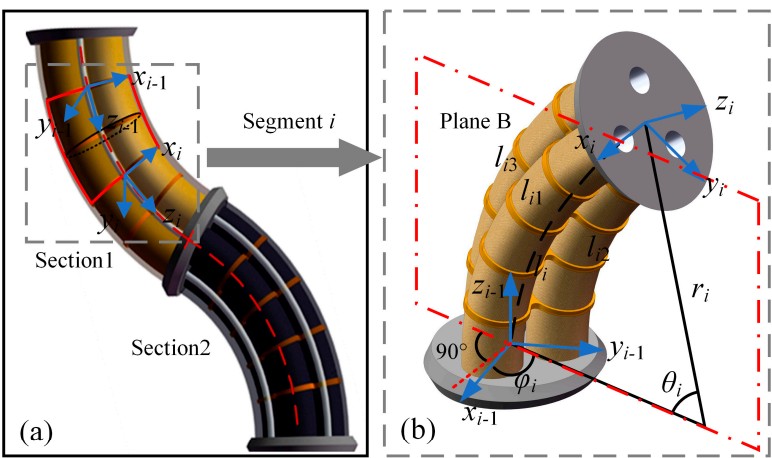

**Figure 1.** (**a**) Discrete segment model of the soft manipulator. (**b**) The shape parameter analysis of the first-segment soft manipulator.

First, we define the actuator space state variable **p**, joint space state variable **g**, and operational space variable **X** as follows:

$$\mathbf{p} = \begin{bmatrix} \mathbf{p}_1 & \cdots & \mathbf{p}_i & \cdots & \mathbf{p}_n \end{bmatrix}^T$$
$$\mathbf{p}_i = \begin{bmatrix} p_{i1} & p_{i2} & p_{i3} \end{bmatrix}^T$$
$$\mathbf{g} = \begin{bmatrix} \mathbf{g}_1 & \cdots & \mathbf{g}_i & \cdots & \mathbf{g}_n \end{bmatrix}^T \tag{1}$$
$$\mathbf{g}_i = \begin{bmatrix} l_i & \theta_i & \varphi_i \end{bmatrix}^T$$
$$\mathbf{X} = \begin{bmatrix} x & y & z \end{bmatrix}^T$$

where pi, j represents the driving pressure of the *j*-th soft unit in the *i*-th segment of the soft manipulator. $i \in \{1, 2 \ldots, n\}$; $j \in \{1, 2, 3\}$.

In our previous work on the dynamic modeling of the soft manipulator, we established a homogeneous transformation matrix between adjacent coordinate systems of the soft manipulator based on the Denavit–Hartenberg method as follows [19]:

$$^{i-1}\mathbf{H}_i = \begin{bmatrix} c_\tau\left(c_\varphi^2(c_\theta - 1) + 1\right) + c_\varphi s_\varphi s_\tau(c_\theta - 1) & c_\varphi s_\varphi c_\tau(c_\theta - 1) - s_\tau\left(c_\varphi^2(c_\theta - 1) + 1\right) & c_\varphi s_\theta & -\frac{l_i c_\varphi(c_\theta - 1)}{\theta_i} \\ s_\tau\left(c_\theta - c_\varphi^2(c_\theta - 1)\right) + c_\varphi s_\varphi c_\tau(c_\theta - 1) & c_\tau\left(c_\theta - c_\varphi^2(c_\theta - 1)\right) - c_\varphi s_\varphi s_\tau(c_\theta - 1) & s_\varphi s_\theta & -\frac{l_i s_\varphi(c_\theta - 1)}{\theta_i} \\ -s_\varphi s_\tau s_\theta - c_\varphi c_\tau s_\theta & c_\varphi s_\tau s_\theta - s_\varphi c_\tau s_\theta & c_\theta & \frac{l_i s_\theta}{\theta_i} \\ 0 & 0 & 0 & 1 \end{bmatrix} \tag{2}$$

$$c_\varphi = \cos \varphi_i \quad c_\theta = \cos \theta_i \quad c_\tau = \cos \tau_i$$
$$s_\varphi = \sin \varphi_i \quad s_\theta = \sin \theta_i \quad s_\tau = \sin \tau_i$$

The mapping relationship from joint space variable **g** to operational space variable **X** can be obtained based on the homogeneous transformation matrix, and we obtain an expression for the end position of the soft manipulator as follows:

$$\begin{bmatrix} \mathbf{X} \\ 1 \end{bmatrix} = \begin{bmatrix} x \\ y \\ z \\ 1 \end{bmatrix} = {}^0\mathbf{H}_1{}^1\mathbf{H}_2 \cdots {}^{n-1}\mathbf{H}_n \begin{bmatrix} 0 \\ 0 \\ 0 \\ 1 \end{bmatrix} \tag{3}$$

By using Equations (1)–(3), the operational space variable **X** can be represented by the joint space variable **g**. This mapping relationship can be represented as $\mathbf{X} = f_{gx}(\mathbf{g})$. We take the derivative of the mapping relation with respect to time to obtain the following:

$$\dot{\mathbf{X}} = \frac{\partial f_{gx}}{\partial \mathbf{g}} \dot{\mathbf{g}} = \mathbf{J}(\mathbf{g})\dot{\mathbf{g}} \tag{4}$$

$$\mathbf{J}(\mathbf{g}) = \frac{\partial f_{gx}}{\partial \mathbf{g}} = \begin{bmatrix} \frac{\partial^0 \mathbf{H}_{n,(1,4)}}{\partial \mathbf{g}_{11}} & \frac{\partial^0 \mathbf{H}_{n,(1,4)}}{\partial \mathbf{g}_{12}} & \cdots & \frac{\partial^0 \mathbf{H}_{n,(1,4)}}{\partial \mathbf{g}_{n3}} \\ \frac{\partial^0 \mathbf{H}_{n,(2,4)}}{\partial \mathbf{g}_{11}} & \frac{\partial^0 \mathbf{H}_{n,(2,4)}}{\partial \mathbf{g}_{12}} & \cdots & \frac{\partial^0 \mathbf{H}_{n,(2,4)}}{\partial \mathbf{g}_{n3}} \\ \frac{\partial^0 \mathbf{H}_{n,(3,4)}}{\partial \mathbf{g}_{11}} & \frac{\partial^0 \mathbf{H}_{n,(3,4)}}{\partial \mathbf{g}_{12}} & \cdots & \frac{\partial^0 \mathbf{H}_{n,(3,4)}}{\partial \mathbf{g}_{n3}} \end{bmatrix} \quad \mathbf{J}(\mathbf{g}) \in \mathbb{R}^{3 \times 3n} \tag{5}$$

According to the pseudo inverse property of the Jacobian matrix, the mapping relationship from operational space variable **X** to joint space variable **g** is obtained: $\dot{\mathbf{g}} = \mathbf{J}^T(\mathbf{J}\mathbf{J}^T)^{-1}\dot{\mathbf{X}}$. Thus, the mapping can be obtained as $\mathbf{g} = f_{xg}(\mathbf{X})$.

### 2.1.2. The Mapping $f_{gp}$ from Joint Space to Actuator Space

The following presents the static force analysis of the soft manipulator. As shown in Figure 2, the soft manipulator is subject to the combined action of gravity, the driving pressure, and elastic force. We analyze the force and moment balance of each segment of the soft manipulator.

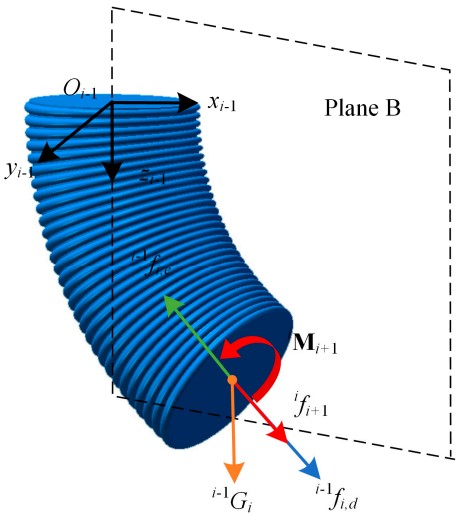

**Figure 2.** Force analysis of the *i*-th segment of the soft manipulator.

Driving Force and Torque: In this paper, a hydraulic drive system is used to drive the soft manipulator. Therefore, the driving force and torque of the soft manipulator are determined by the driving pressure inside the soft unit. In the local coordinate system, the direction of the driving pressure is always perpendicular to the cross-section of the soft manipulator. The driving torque is generated by the driving pressure. Therefore, the driving force and driving torque of the *i*-th segment of the soft manipulator in its local coordinate system are as follows:

$$^{i-1}f_{i,d} = \sum_{j=1}^{3} p_{i,j} A_d$$

$$^{i-1}\mathbf{M}_{i,d} = \sum_{j=1}^{3} \left( \mathbf{r}_{d,j} \times \begin{bmatrix} 0 \\ 0 \\ p_{i,j} A_d \end{bmatrix} \right) \qquad \mathbf{r}_{d,j} \in \mathbb{R}^{3 \times 1} \tag{6}$$

where $p_{i,j}$ represents the driving pressure of the *j*-th soft unit in the *i*-th segment of the soft manipulator; $A_d$ represents the effective area inside the soft unit, and $\mathbf{r}_{d,j}$ represents the vector diameter of the *j*-th soft unit.

Elastic Force and Torque: Firstly, we make the following assumptions about soft materials: (1) soft materials have isotropy; (2) the total volume of soft materials remains unchanged before and after deformation; and (3) Hookean law can be used to calculate the elastic force of soft materials, and Hookean law can still be used in large deformation. During the simulation and experimental process, we will set a limit on the driving force of the soft manipulator to ensure that the soft material will not undergo significant deformation. For soft materials, the elastic force and torque are only related to the degree of deformation. According to the elastic deformation formula, the elastic force and torque of the soft manipulator can be obtained as follows:

$$^{i-1}f_{i,e} = EA_s \sum_{j=1}^{3} \frac{(l_{ij} - l_{ij0})}{l_{ij0}} = 3EA_s \frac{l_i - l_{i0}}{l_{i0}}$$

$$^{i-1}\mathbf{M}_{i,e} = \mathbf{K}_i \begin{bmatrix} \frac{\theta_i \cos \varphi_i}{l_i} & \frac{\theta_i \sin \varphi_i}{l_i} & 0 \end{bmatrix}^{\mathrm{T}} \tag{7}$$

$$\mathbf{K}_i = diag([EI_{xxi} \ EI_{yyi} \ 0])$$

where $E$ is the elastic modulus of the soft unit; $A_s$ is the annular solid area of the soft unit; $l_{ij0}$ is the initial length of the *j*-th soft unit in the *i*-th segment of the soft manipulator;

$I_{xxi} = I_{yyi} = \pi r_i^4/4$ represents the sectional moment of inertia in the x and y directions, respectively; $r_i$ represents the radius of the soft manipulator.

Gravity and Torque: In this paper, the lumped mass method is used to approximate the mass of each segment of the soft manipulator based on the lumped mass point acting on its end, and the direction of gravity is always vertically downward. We assume that the gravity of the entire soft manipulator is $G$, so the vector of the gravity of each segment of the soft manipulator in the base coordinate system is expressed as $[0\ 0\ G/n]^T$. Therefore, the expressions for the gravity force and moment of each segment of the soft manipulator are as follows:

$$
\begin{aligned}
^{i-1}f_{i,G} &= [0\ \ 0\ \ 1\ \ 0]\,(^0\mathbf{H}_i)^{-1}[0\ \ 0\ \ \tfrac{G}{n}\ \ 0]^T \\
^{i-1}\mathbf{M}_{i,G} &= {}^{i-1}\boldsymbol{\gamma}_i \times {}^{i-1}\mathbf{F}_{i,G} \quad {}^{i-1}\boldsymbol{\gamma}_i \in \mathbb{R}^{3\times1}
\end{aligned}
\tag{8}
$$

$$
^{i-1}\mathbf{F}_{i,G} =
\begin{bmatrix}
1 & 0 & 0 & 0 \\
0 & 1 & 0 & 0 \\
0 & 0 & 1 & 0
\end{bmatrix}
(^0\mathbf{H}_i)^{-1}[0\ \ 0\ \ \frac{G}{n}\ \ 0]^T
\tag{9}
$$

where $^{i-1}\boldsymbol{\gamma}_i$ is the position coordinate of the mass point of the *i*-th segment of the soft manipulator under the current local coordinate system.

Force and Moment Balance: Because the soft manipulator is made of elastic material, its force at both ends is transmitted through the elastic strain of the elastomer. In a mechanical analysis of a segment of the soft manipulator, it is necessary to consider the influence of adjacent segments. Therefore, the force and moment balance equations of the *i*-th segment of the soft manipulator are as follows:

$$
\begin{aligned}
^{i-1}f_{i,d} &= {}^{i-1}f_{i,e} - {}^{i-1}f_{i,G} - {}^if_{i+1,e} + {}^if_{i+1,d} \\
\mathbf{M}_{i,d} &= {}^{i-1}\mathbf{M}_{i,e} - {}^{i-1}\mathbf{M}_{i,G} - {}^i\mathbf{M}_{i+1,e} + {}^i\mathbf{M}_{i+1,d}
\end{aligned}
\tag{10}
$$

The mapping from the joint space variable **g** to the actuator space variable **p** is derived from the force and torque balance relationship of the above soft manipulator as follows: $\mathbf{p} = f_{gp}(\mathbf{g})$. Therefore, the mapping from operational space variable **X** to actuator space variable **p** can be obtained as $\mathbf{p} = f_{gp}(f_{xg}(\mathbf{X}))$. Through this mapping relationship, the driving pressure can be obtained from the desired end position of the soft manipulator.

### 2.2. Static Feedforward Model with Contact Constraints

During the application process, the soft manipulator will be affected by environmental obstacles, which will make it unable to follow the intended track. Therefore, we improved the original model by establishing a static feedforward model with contact constraints. The operational state of the soft manipulator can be divided into noncontact and contact states. The noncontact state signifies that the soft manipulator is not affected by the obstacle during the movement. The contact state signifies that the soft manipulator contacts the obstacle before reaching the desired position and encounters a nonzero contact force. In order to detect the operational state of the soft manipulator, we designed a contact detection process as shown in Figure 3. First, we judge the operation state of the soft manipulator. We input the desired end point position of the soft manipulator and the position of the obstacle. Then, we calculate the corresponding geometric path of the soft manipulator based on the noncontact feedforward static model and calculate the motion space area$_1$ comprising the soft manipulator path from its initial state to its final state. Finally, the operation state of the soft manipulator can be determined by checking whether the obstacle space area$_2$ is within the moving space area$_1$ of the soft manipulator.

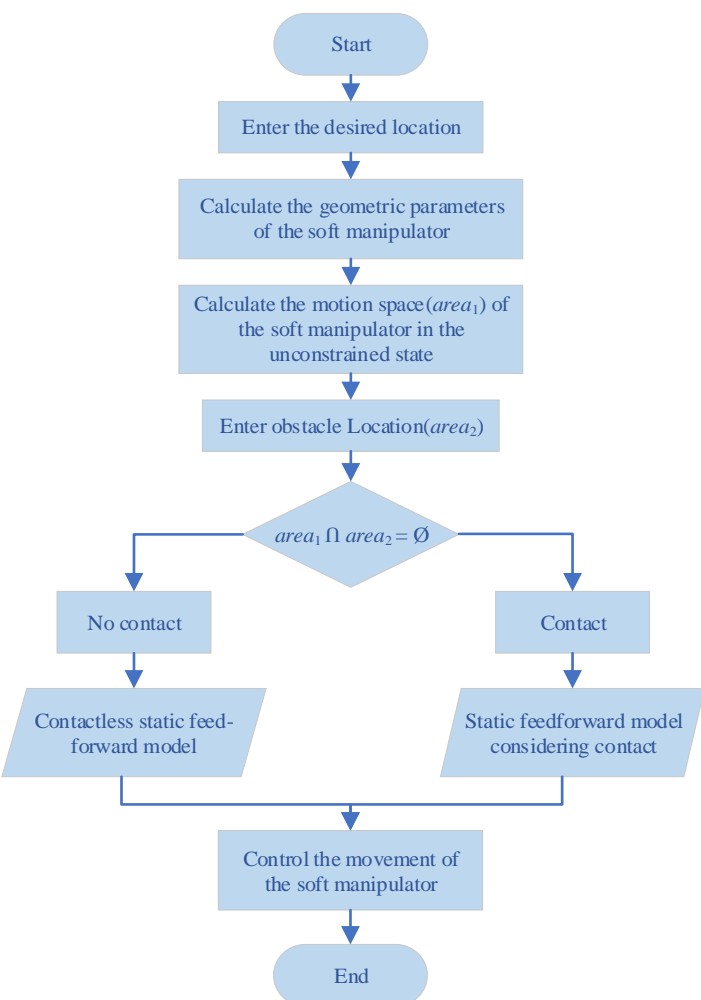

**Figure 3.** Operational state detection process of soft manipulator.

After judging the operational state of the soft manipulator, if there is no contact between the soft manipulator and the obstacle, the static feedforward model is used to control the motion of the soft manipulator. If there is contact between the soft manipulator and the obstacle, the control accuracy of the original static feedforward model is significantly reduced. Therefore, we improve the original static feedforward model. In its original state, we divide the contact-discrete manipulator segment into two continuous arcs with different curvatures, as shown in Figure 4. Because the soft manipulator has infinite degrees of freedom, any contact point can affect its final position. When the $i$-th segment of the soft manipulator generates c contact points, this $i$-th segment can be divided into $c + 1$ discrete segments. We simplify this model for ease of calculation. We assume that the soft manipulator has only one contact point with the environment, and the contact position is $\mathbf{X}_{j1} = [x_{j1}, y_{j1}, z_{j1}, 1]^T$. The soft manipulator is divided into a contact-front segment and a contact-rear segment. When establishing the static feedforward model of the soft manipulator with contact constraints, the contact-front and contact-rear segments are analyzed. First, we calculate the equivalent contact point position along the central axis of the soft manipulator. After contact collision, the contact point on the soft manipulator and the environmental obstacle point are in the same position in space. We describe this condition as follows:

$$^0\mathbf{H}_a\left(r_s \cos \varphi_a \quad r_s \sin \varphi_a \ 0 \ 1\right)^{\mathrm{T}} = \mathbf{X}_{j1} \tag{11}$$

where $r_s$ is the cross-section radius of the soft manipulator, and $\varphi_a$ represents the deflection angle of the $a$-th-segment soft manipulator.

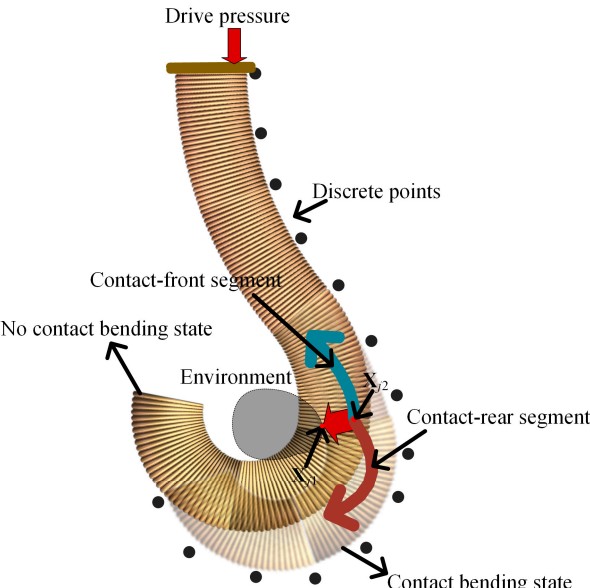

**Figure 4.** The motion state of the soft manipulator after contact with the environment.

According to the above equation, when the contact point position is known, the homogeneous change matrix $^0\mathbf{H}_a$ of the soft manipulator contact-front segment can be obtained, and the equivalent contact point position is $\mathbf{X}_{j2} = [x_{j2}, y_{j2}, z_{j2}, 1]^T$. Because this contact point position is fixed, the equivalent contact point position is regarded as the expected position of the end of the contact-front segment. We define the desired position of the end point of the entire soft manipulator as $\mathbf{X} = [x, y, z, 1]^T$, such that

$$^0\mathbf{H}_a\mathbf{X}_{j3} = \mathbf{X} \tag{12}$$

where $\mathbf{X}_{j3}$ represents the expected position of the end point of the contact rear segment of the soft manipulator.

The expected position $\mathbf{X}_{j3}$ of the end point of the contact-rear segment can also be obtained from the above formula. The driving pressure can then be obtained by bringing the expected positions of the contact-front and contact-rear segments into the upper section of the static feedforward model.

## 3. Design Controller

In the previous section, we established the static feedforward model of the soft manipulator and showed how the control of the soft manipulator can be realized with the model. In practical applications, the environment will interfere with the soft manipulator. In these cases, the control task of the soft manipulator cannot be completed by exclusively using feedforward control. Therefore, we design a sliding mode controller based on the operational space dynamics model and combine this controller with the feedforward compensation model to realize position control of the soft manipulator.

### 3.1. Operational Space Dynamics

In previous work, we established a dynamic model of the soft manipulator based on the Newton–Euler iterative method [19]. Based on this dynamic model, the dynamic equation of the soft manipulator in joint space can be obtained as follows:

$$\mathbf{M}(\mathbf{g})\ddot{\mathbf{g}} + \mathbf{C}(\mathbf{g}, \dot{\mathbf{g}})\dot{\mathbf{g}} + \mathbf{N}(\mathbf{g}) = \boldsymbol{\tau}(\mathbf{p}) \tag{13}$$

where $\mathbf{M}(\mathbf{g}) \in \mathbb{R}^{3n \times 3n}$ is the inertial force matrix; $\mathbf{C}(\mathbf{g}, \dot{\mathbf{g}}) \in \mathbb{R}^{3n \times 3n}$ is the Coriolis force and damping force matrix; $\mathbf{N}(\mathbf{g}) \in \mathbb{R}^{3n \times 1}$ is a composite matrix, including gravity, elastic

force, and the interaction between adjacent soft units; and $\boldsymbol{\tau}(\mathbf{p}) \in \mathbb{R}^{3n \times 1}$ represents the driving force.

In Section 2, we define the end position of the soft manipulator as $\mathbf{X} = [x, y, z]^T$ and then obtain a mapping between operational space variable $\mathbf{X}$ and joint space variable $\mathbf{g}$: $\mathbf{X} = f_{gx}(\mathbf{g})$. We calculate the first and second derivatives of the mapping with respect to time to obtain

$$
\begin{aligned}
\dot{\mathbf{X}} &= \mathbf{J}(\mathbf{g})\dot{\mathbf{g}} \qquad \mathbf{J}(\mathbf{g}) \in \mathbb{R}^{3 \times 3n} \\
\ddot{\mathbf{X}} &= \mathbf{J}(\mathbf{g})\ddot{\mathbf{g}} + \frac{d(\mathbf{J}(\mathbf{g}))}{dt}\dot{\mathbf{g}}
\end{aligned}
\tag{14}
$$

where $\mathbf{J}(\mathbf{g})$ is the Jacobian matrix associated with the operational space.

We further combine (13)–(14) to obtain

$$
\ddot{\mathbf{X}} = \mathbf{J}\mathbf{M}^{-1}\boldsymbol{\tau} - \mathbf{J}\mathbf{M}^{-1}\mathbf{C}\dot{\mathbf{g}} - \mathbf{J}\mathbf{M}^{-1}\mathbf{N} + \frac{d(\mathbf{J})}{dt}\dot{\mathbf{g}}
\tag{15}
$$

According to the operational space dynamics model in Reference [30], we rewrite the matrix in the joint space dynamics model of the soft manipulator as follows:

$$
\begin{aligned}
\boldsymbol{\tau} &= \mathbf{J}^T\mathbf{f} \qquad \mathbf{f} \in \mathbb{R}^{3 \times 1} \\
\mathbf{M}_p &= \left(\mathbf{J}\mathbf{M}^{-1}\mathbf{J}^T\right)^{-1} \qquad \mathbf{M}_p \in \mathbb{R}^{3 \times 3} \\
\mathbf{C}_p &= \mathbf{M}_p\left[\mathbf{J}\mathbf{M}^{-1}\mathbf{C}\dot{\mathbf{g}} - \frac{d(\mathbf{J})}{dt}\dot{\mathbf{g}}\right] \qquad \mathbf{C}_p \in \mathbb{R}^{3 \times 1} \\
\mathbf{N}_p &= \mathbf{M}_p\mathbf{J}\mathbf{M}^{-1}\mathbf{N} \qquad \mathbf{N}_p \in \mathbb{R}^{3 \times 1}
\end{aligned}
\tag{16}
$$

By combining (13), (15), and (16), the dynamic equation of the operational space of the soft manipulator can be given as follows:

$$
\mathbf{M}_p\ddot{\mathbf{X}} + \mathbf{C}_p + \mathbf{N}_p = \mathbf{f}
\tag{17}
$$

### 3.2. Design Sliding Mode Controller

First, we assume that there is an interference effect in the operational space and define it as an external disturbance force $\boldsymbol{\Delta}$. Then the dynamic model for the operational space of the soft manipulator is rewritten as

$$
\mathbf{M}_p\ddot{\mathbf{X}} + \mathbf{C}_p + \mathbf{N}_p + \boldsymbol{\Delta} = \mathbf{f}
\tag{18}
$$

In this operational space, the system error is defined as $\mathbf{e} = \mathbf{X} - \mathbf{X}_d$, and $\mathbf{X}_d$ is the expected value of the end position. According to the sliding mode control method, we set the sliding mode surface as $\mathbf{s} = c\mathbf{e} + \dot{\mathbf{e}}, (c > 0)$ and define the sliding mode surface approach rate as an exponential rate:

$$
\dot{\mathbf{s}} = -\varepsilon \mathrm{sgn}(\mathbf{s}) - k\mathbf{s} \qquad \varepsilon > 0, k > 0
\tag{19}
$$

In addition, we take the derivative of the sliding mode surface with respect to time to obtain

$$
\dot{\mathbf{s}} = c\dot{\mathbf{e}} + \ddot{\mathbf{e}} = c\dot{\mathbf{e}} + \ddot{\mathbf{X}} - \ddot{\mathbf{X}}_d
\tag{20}
$$

It can be obtained by combining the (19)–(20):

$$
\ddot{\mathbf{X}} = -c\dot{\mathbf{e}} + \ddot{\mathbf{X}}_d - \varepsilon \mathrm{sgn}(\mathbf{s}) - k\mathbf{s}
\tag{21}
$$

The control law can be obtained by introducing (21) into (18):

$$
\boldsymbol{\tau} = \mathbf{J}^T\mathbf{f} = \mathbf{J}^T(\mathbf{M}_p(-c\dot{\mathbf{e}} + \ddot{\mathbf{X}}_d - \varepsilon \mathrm{sgn}(\mathbf{s}) - k\mathbf{s}) + \mathbf{C}_p + \mathbf{N}_p + \boldsymbol{\Delta})
\tag{22}
$$

In this control law, because $\boldsymbol{\Delta}$ is unknown, we consider an estimated value $\boldsymbol{\Delta}_e$ instead of $\boldsymbol{\Delta}$. We consider the upper bound of the interference force as $\boldsymbol{\Delta}_p$. During the controller verification process, we set it to $\Delta_p = 5\,N$. And this estimated value is defined as $\boldsymbol{\Delta}_e = -\boldsymbol{\Delta}_p \cdot \text{sgn}(\mathbf{s})$. Thus, the control law becomes

$$\boldsymbol{\tau} = \mathbf{J}^T(\mathbf{M}_p(-c\dot{\mathbf{e}} + \ddot{\mathbf{X}}_d - \varepsilon\text{sgn}(\mathbf{s}) - k\mathbf{s}) + \mathbf{C}_p + \mathbf{N}_p - \boldsymbol{\Delta}_p\text{sgn}(\mathbf{s})) \tag{23}$$

We substitute the modified control law back to obtain the modified sliding mode surface approach rate as follows:

$$\dot{\mathbf{s}} = -\varepsilon\text{sgn}(\mathbf{s}) - k\mathbf{s} + (\mathbf{M}_p^{-1}(-\Delta_p\text{sgn}(\mathbf{s}) - \boldsymbol{\Delta})) \tag{24}$$

With the controller designed, we use the second law of Lyapunov to judge whether it meets stability requirements. The Lyapunov function is defined as $V = \mathbf{s}^T\mathbf{s}$, $V > 0$. It can then be proven that

$$
\begin{aligned}
\dot{V} &= \dot{\mathbf{s}}^T\mathbf{s} + \mathbf{s}^T\dot{\mathbf{s}} \\
&= -\varepsilon\,\text{sgn}(\mathbf{s}^T)\mathbf{s} - k\mathbf{s}^T\mathbf{s} + (\mathbf{M}_p^{-1}(-\Delta_p\text{sgn}(\mathbf{s}) - \boldsymbol{\Delta}))^T\mathbf{s} - \varepsilon\mathbf{s}^T\text{sgn}(\mathbf{s}) - k\mathbf{s}^T\mathbf{s} + \mathbf{s}^T\mathbf{M}_p^{-1}(-\Delta_p\text{sgn}(\mathbf{s}) - \boldsymbol{\Delta}) \\
&= -\varepsilon\,\text{sgn}(\mathbf{s}^T)\mathbf{s} - k\mathbf{s}^T\mathbf{s} + (\mathbf{M}_p^{-1}(-\Delta_p - \tfrac{\boldsymbol{\Delta}}{\text{sgn}(\mathbf{s})}))^T\text{sgn}(\mathbf{s})^T\mathbf{s} - \varepsilon\mathbf{s}^T\text{sgn}(\mathbf{s}) - k\mathbf{s}^T\mathbf{s} + \mathbf{s}^T\text{sgn}(\mathbf{s})\mathbf{M}_p^{-1}(-\Delta_p - \tfrac{\boldsymbol{\Delta}}{\text{sgn}(\mathbf{s})})
\end{aligned}
\tag{25}
$$

Because $\mathbf{s}^T\mathbf{s} > 0$, $\text{sgn}(\mathbf{s}^T)\mathbf{s} > 0$, $\mathbf{s}^T\text{sgn}(\mathbf{s}) > 0$, $\text{sgn}(\mathbf{s})^T\mathbf{s} > 0$, $\Delta_p > \Delta$, and $\mathbf{M}_p$ is a positive definite matrix, we can obtain

$$\dot{V} < \left(\mathbf{M}_p^{-1}\left(-\Delta_p - \frac{\boldsymbol{\Delta}}{\text{sgn}(\mathbf{s})}\right)\right)^T\text{sgn}(\mathbf{s})^T\mathbf{s} + \mathbf{s}^T\text{sgn}(\mathbf{s})\mathbf{M}_p^{-1}\left(-\Delta_p - \frac{\boldsymbol{\Delta}}{\text{sgn}(\mathbf{s})}\right) < 0 \tag{26}$$

In summary, the Lyapunov function yields $V > 0$ and $\dot{V} < 0$, indicating that the designed controller meets the stability requirements. Figure 5 shows the control block diagram of the soft manipulator. The feedforward compensation model is combined with the sliding mode controller to execute position control of the soft manipulator.

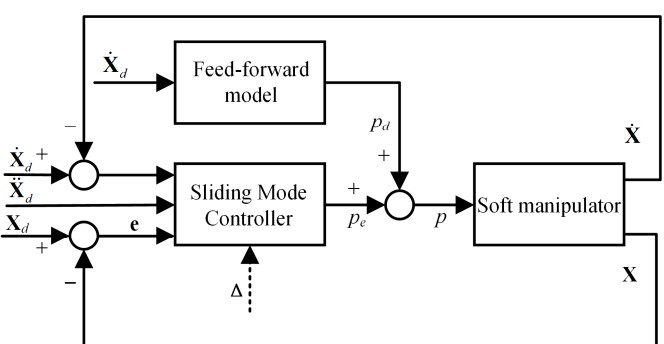

**Figure 5.** Control block diagram of combined action of feedforward and feedback.

## 4. Simulation and Experiment

### 4.1. Simulation Result

We use the feedforward compensation model and sliding mode controller together. The parameters of the soft manipulator and the sliding mode controller are shown in Table 1. Firstly, we define the positional errors during the simulation and experimental processes. The absolute position error represents the difference between the actual measurement position of the end point of the soft manipulator in that direction and the desired position. The relative position error represents the ratio of absolute position error to the desired position of the soft manipulator in that direction.

**Table 1.** Relevant parameters.

| Symbol | Variable Name | Value |
|--------|---------------|-------|
| $E$ | the elastic modulus of the soft unit | 0.6 MPa |
| $A_d$ | the effective area inside the soft unit | $3.14 \times 10^{-4}$ m$^2$ |
| $A_s$ | the annular solid area of the soft unit | $3.93 \times 10^{-4}$ m$^2$ |
| $r_s$ | the cross-section radius of the soft manipulator | 0.05 m |
| $c$ | the coefficient of the sliding mode surface | 3 |
| $k$ | the coefficient of the sliding mode surface approach rate | 10 |
| $\varepsilon$ | the coefficient of the sliding mode surface approach rate | 0.01 |
| $\Delta_p$ | the upper bound of the interference force | 5 N |
| $G$ | the gravity of the entire soft manipulator | 24.5 N |

In order to improve simulation efficiency and verify the control accuracy of the controller, the entire soft manipulator is divided into two constant curvature arcs (i.e., $n = 2$) when designing the feedforward controller and the sliding mode controller. In the simulation process, we use the previously established dynamic model of the soft manipulator as the controlled object [19], and divide the controlled soft manipulator into four constant curvature arcs (i.e., $n = 4$). In addition, we observed the curvature components in the $x$ and $y$ directions of each segment of the soft manipulator as follows:

$$
\begin{aligned}
kx_i &= \frac{\theta_i \cos \varphi_i}{l_i} \\
ky_i &= \frac{\theta_i \sin \varphi_i}{l_i}
\end{aligned}
\tag{27}
$$

where $kx_i$ represents the curvature component of the $i$-th segment of the soft manipulator in the $x$ direction, and $ky_i$ represents the curvature component of the $i$-th segment of the soft manipulator in the $y$ direction.

First, we control the motion of the soft manipulator in the $xoz$ plane. We assign a desired position for the end point of the soft manipulator and use the static feedforward model to move the soft manipulator into this desired position. At the same time, we record the motion trajectory, curvature, measured position, desired position, and position error of the soft manipulator, respectively. The results are shown in Figure 6. In Figure 6, we can find that the end point of the soft manipulator fails to reach the desired point. The average absolute error of the end point of the soft manipulator in the $x$ direction is 0.028 m, the peak absolute error is 0.035 m, and the relative errors are 17.6% and 22%, respectively. The average absolute error of the end point of the soft manipulator in the $z$ direction is 0.018 m, the peak absolute error is 0.021 m, and the relative errors are 3.5% and 4%, respectively. Through analysis, it can be found that the position error is large when using the static feedforward model to control the movement of the soft manipulator. The reason is that when designing the feedforward model controller, we approximate each segment of the soft manipulator as an arc with constant curvature. However, the soft manipulator is affected by gravity and other factors during its movement, such that the deformation of the soft manipulator does not yield an arc with constant curvature. Therefore, it is relatively difficult to accurately control the soft manipulator to reach the desired position when exclusively using the feedforward controller. We thus design a sliding mode controller based on the operational space dynamics model. We use the feedforward compensation model and sliding mode controller together for simulation analysis, and the results are shown in Figure 7. In Figure 7, we show that the tracking accuracy of the end point of the soft manipulator is superior to the previous method. The average absolute error of the end point of the soft manipulator in the $x$ direction is 0.0024 m, the peak absolute error is 0.0064 m, and the relative errors are 1.5% and 4%, respectively. The average absolute error of the end point of the soft manipulator in the $z$ direction is 0.0054 m, the peak absolute error is 0.0081 m, and the relative errors are 1% and 1.5%, respectively. Through analysis, it can be found that the position error is very small when the static feedforward model and the sliding mode controller are used to jointly control the movement of the

soft manipulator. However, in practical applications, the soft manipulator will be affected by environmental interference. We add an interference force to the dynamic model of the controlled soft manipulator. As shown in Figure 8a, the interference force acts on the center of the entire soft manipulator along the *x*-axis direction. The interference force should be smaller than the upper bound $\Delta_p$ of the interference force proposed in Section 3, so we set the interference force *f* to 4 *N*. The results are shown in Figure 8. It can be seen in Figure 8a that even if the soft manipulator is affected by an interference force, the controller can still move the soft manipulator in the desired direction and reach the desired position. It can be seen in Figure 8d that the average absolute error of the end point of the soft manipulator in the *x* direction is 0.008 m, the peak absolute error is 0.056 m, and the relative errors are 5% and 35%, respectively. The average absolute error of the end point of the soft manipulator in the *z* direction is 0.0081 m, the peak absolute error is 0.016 m, and the relative errors are 1.5% and 3%, respectively. Through analysis, it can be found that when the interference force first acts, it causes significant positional errors in the soft manipulator, but the positional errors quickly decrease and tend to stabilize. The analysis of Figures 7 and 8 shows that the designed controller has good anti-interference capability and the control strategy with the combination of feedforward and feedback has high accuracy in performing tasks.

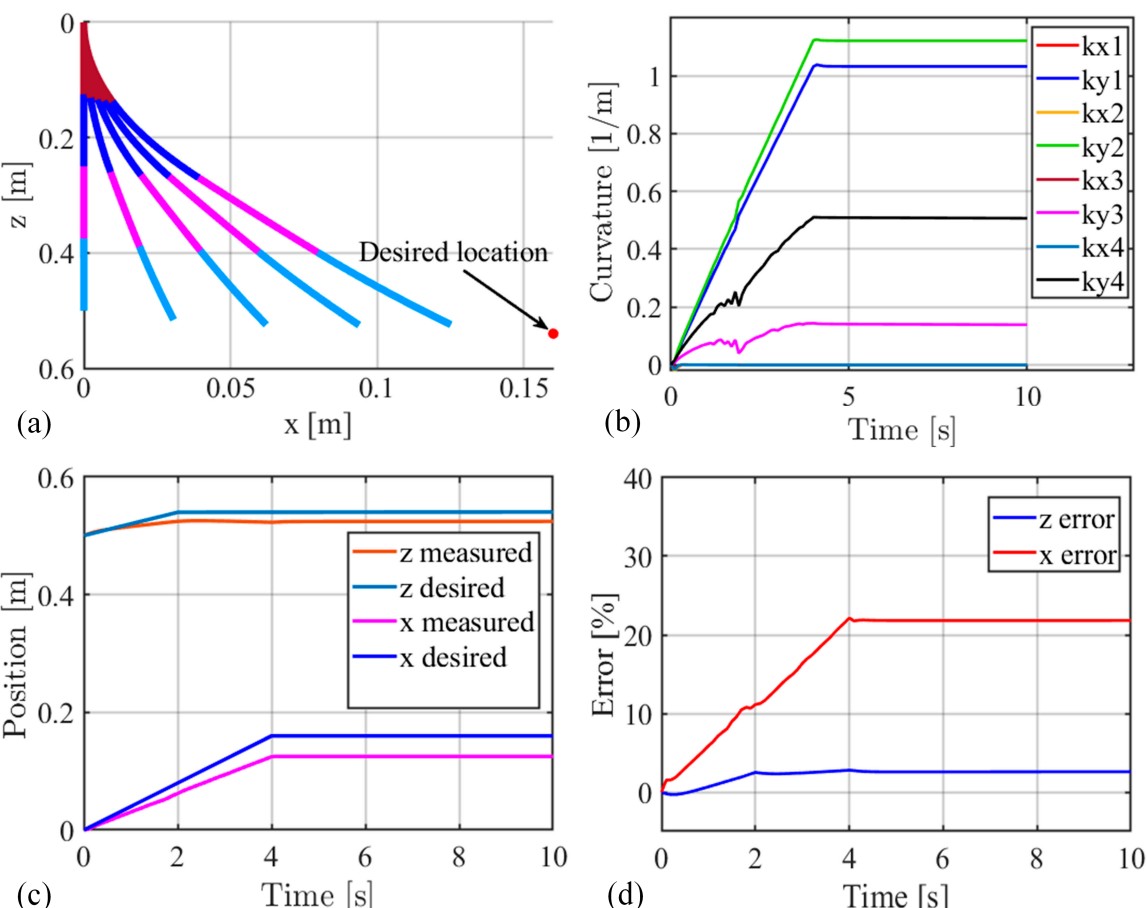

**Figure 6.** Only the feedforward model is used to control the soft manipulator to reach the desired position. (**a**) The trajectory of the soft manipulator. (**b**) The curvature of the soft manipulator. (**c**) The desired position and measured position of the end point of the soft manipulator. (**d**) The position error of the end point of the soft manipulator.

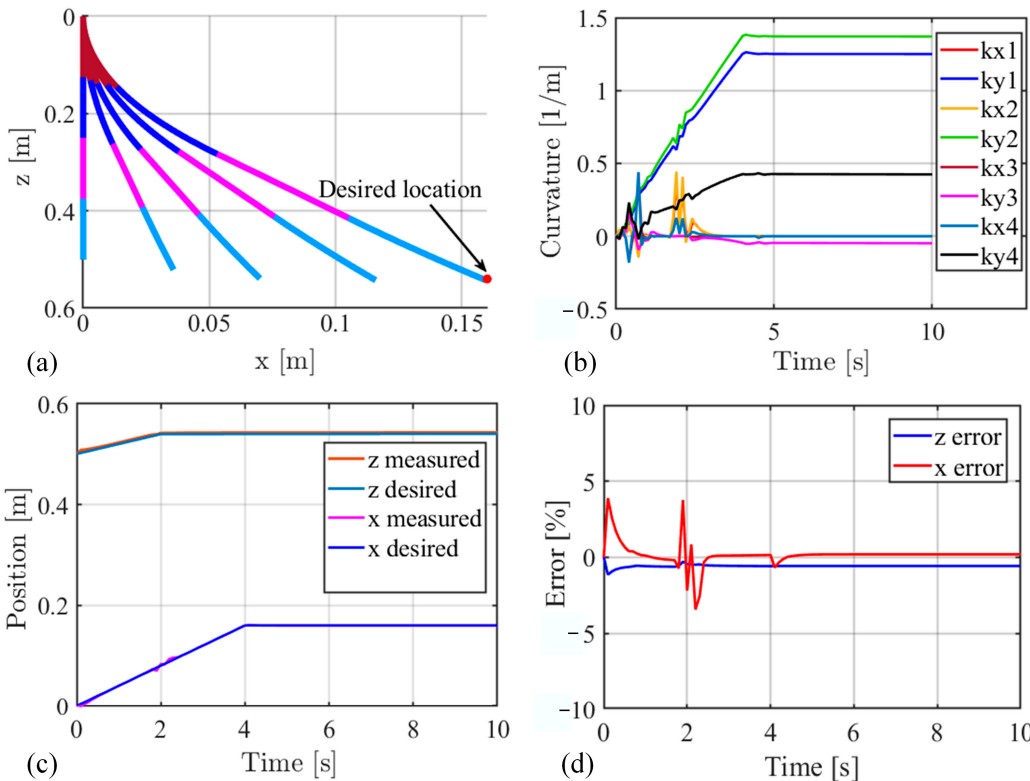

**Figure 7.** The soft manipulator is controlled to reach the desired position by combining feedforward and feedback. (**a**) The trajectory of the soft manipulator. (**b**) The curvature of the soft manipulator. (**c**) The desired position and measured position of the end point of the soft manipulator. (**d**) The position error of the end point of the soft manipulator.

In addition, we verify the accuracy of the controller in three-dimensional space by controlling the soft manipulator to perform circular motion. The simulation results are shown in Figure 9, where we give the expected trajectory of the end point of the soft manipulator. First, we move from initial position 1 to position 2, then move circularly in a counterclockwise direction to pass through positions 3, 4, and 5, before finally returning to position 2. By analyzing Figure 9a,c, it can be seen that the soft manipulator model performs circular motion in three-dimensional space that is consistent with the expected trajectory. It can be seen in Figure 9d that the average absolute error of the end point of the soft manipulator in the $x$ direction is 0.0098 m, the peak absolute error is 0.0201 m, and the relative errors are 6.1% and 13%, respectively. The average absolute error of the end point of the soft manipulator in the $z$ direction is 0.0051 m, the peak absolute error is 0.0093 m, and the relative errors are 3.2% and 5.8%, respectively. Through analysis, it can be found that the instantaneous error between the end position of the soft manipulator and the desired position is very small. From the above simulation, it can be seen that the sliding mode controller with feedforward compensation has good control accuracy for the simulated tasks.

### 4.2. Experimental Result

To verify the performance of the controller in practical applications, we build an experimental platform. A flow chart for the experimental system is shown in Figure 10. The length of the entire soft manipulator is 0.56 m, and the weight is 2.5 kg. The soft manipulator is composed of two sections. Each section contains three soft units, and a rubber hose passes through the soft manipulator to drive each soft unit. These soft units are made of silica gel and coated with elastic fabric to limit their radial expansion. We affix a mark point at the end of the soft manipulator and complete motion capture by identifying

this mark point using a camera. The camera transmits the captured mark point position and speed to an upper computer. A controller in the upper computer calculates the required driving pressure using the position and speed data. A drive pressure is output to the drive system to control the motion of the soft manipulator, thus realizing position control. To more simply observe the motion of the soft manipulator, this position control experiment is conducted in the *xoz* plane.

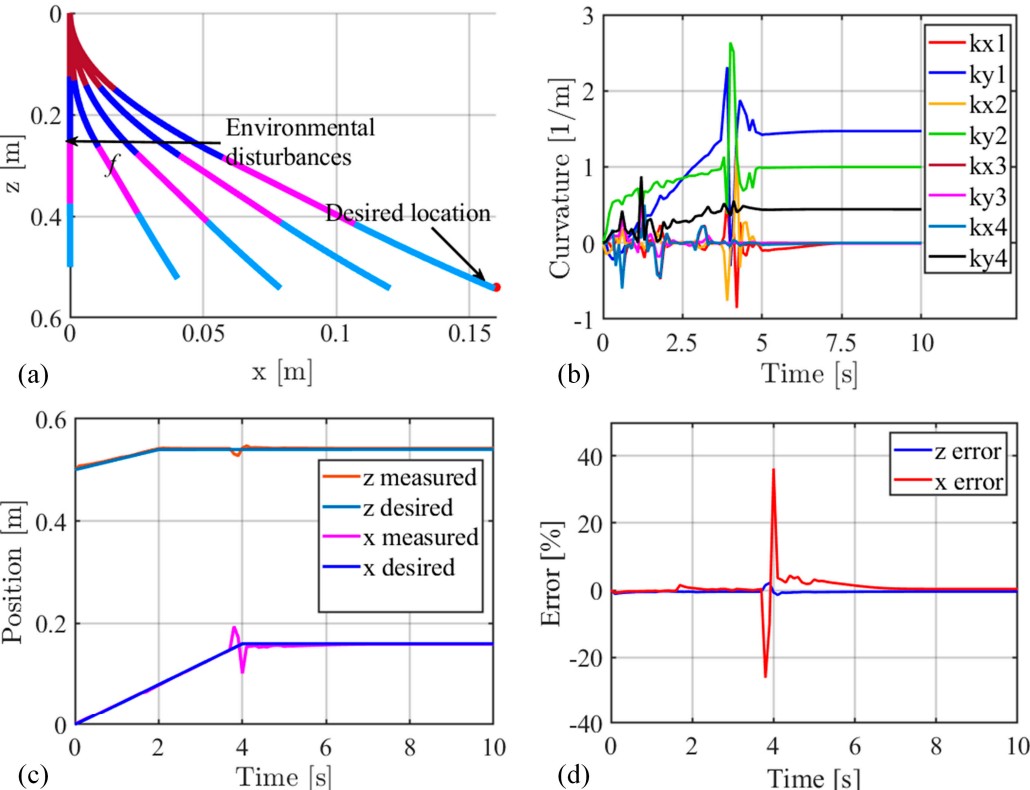

**Figure 8.** When considering the environmental disturbance, the joint action of feedforward and feedback is used to control the soft manipulator to reach the desired position. (**a**) The trajectory of the soft manipulator. (**b**) The curvature of the soft manipulator. (**c**) The desired position and measured position of the end point of the soft manipulator. (**d**) The position error of the end point of the soft manipulator.

### 4.2.1. Trajectory-Tracking Experiment

We carry out trajectory-tracking experiments to verify the performance of the controller. First, we set the desired trajectory of the end point of the soft manipulator: from (0, 0.56) to (0.1, 0.57) to (0.2, 0.53). The experimental results are shown in Figure 11. It can be seen in Figure 11 that the average absolute error of the end point of the soft manipulator in the *x* direction is 0.025 m, the peak absolute error is 0.04 m, and the relative errors are 12.5% and 20%, respectively. The average absolute error of the end point of the soft manipulator in the *z* direction is 0.009 m, the peak absolute error is 0.018 m, and the relative errors are 1.7% and 3.4%, respectively. In addition, we control the soft manipulator to carry out a trajectory-tracking experiment by performing a left–right swing through the plane. We set the desired track of the end point of the soft manipulator to move from (0, 0.56) to (0.2, 0.53) to (−0.2, 0.53) and then to return to the starting point. The experimental results are shown in Figure 12. It can be seen from Figure 12 that the average absolute error of the end point of the soft manipulator in the *x* direction is 0.021 m, the peak absolute error is 0.0404 m, and the relative errors are 10.5% and 20.2%, respectively. The average absolute error of the end point of the soft manipulator in the *z* direction is 0.0098 m, the peak absolute error is 0.02 m, and the relative errors are 1.8% and 3.8%, respectively. Comparing and analyzing the data

in Figures 11 and 12 shows that the position error of the soft manipulator is relatively large at the initial condition when the motion state changes. Due to the presence of air inside the hydraulic cylinder and soft units, there will be a certain hysteresis effect when we drive the soft manipulator to move. This hysteresis effect can cause significant positional errors in the initial motion of the soft manipulator. In addition, we found that the absolute error in the $z$ direction is small, but this relative error is large, which we attribute to the total displacement of the soft manipulator in the $z$ direction being small. In this case, the small displacement fluctuation produces a large relative error. This relative error decreases with increasing total displacement in the $z$ direction. The above analysis demonstrates that the controller designed in this paper has good trajectory-tracking performance.

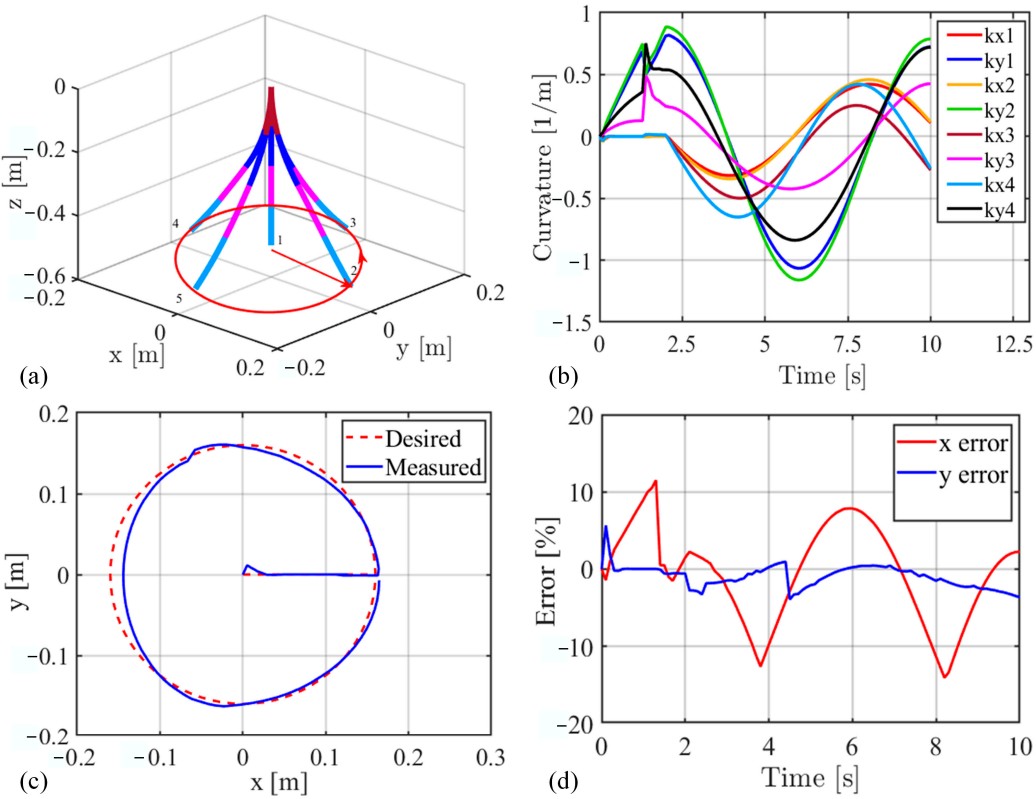

**Figure 9.** Under the action of the controller, the soft manipulator is controlled to move circularly in 3D space. (**a**) The trajectory of the soft manipulator. (**b**) The curvature of the soft manipulator. (**c**) The desired position and measured position of the end point of the soft manipulator. (**d**) The position error of the end point of the soft manipulator.

#### 4.2.2. Collision Experiment

To verify the controller performance in a constrained environment, we conduct a collision experiment. First, we use the static feedforward model without contact constraints and a sliding mode controller to jointly control the soft manipulator for collision experiments. We set the desired trajectory of the end point of the soft manipulator: from the initial point to $(-0.2, 0.53)$. The experimental results are shown in Figure 13. It can be seen from Figure 13 that the average absolute error of the end point of the soft manipulator in the $x$ direction is 0.038 m, the peak absolute error is 0.073 m, and the relative errors are 19% and 36.5%, respectively. The average absolute error of the end point of the soft manipulator in the $z$ direction is 0.0201 m, the peak absolute error is 0.0263 m, and the relative errors are 3.8% and 5%, respectively. We find that the position error of the end point of the soft manipulator increases in the collision experiment, which indicates that environmental constraints have a negative impact on the controller performance. Therefore, we improve the original static feedforward model. A static feedforward model with contact constraints

is established and applied to the collision control experiment. The desired trajectory is set to move the manipulator from the initial point to $(-0.2, 0.53)$. The experimental results are shown in Figure 14. It can be seen from Figure 14 that the average absolute error of the end point of the soft manipulator in the $x$ direction is 0.0277 m, the peak absolute error is 0.0617 m, and the relative errors are 13.8% and 30.8%, respectively. The average absolute error of the end point of the soft manipulator in the $z$ direction is 0.0095 m, the peak absolute error is 0.021 m, and the relative errors are 1.8% and 3.9%, respectively. Comparing Figures 13 and 14 shows that when the improved static feedforward model is used for collision control experiments, the error is significantly reduced between the expected position of the end point of the soft manipulator and the measured position. The above analysis shows that the controller using the improved static feedforward compensation model has better control performance in the constrained environment.

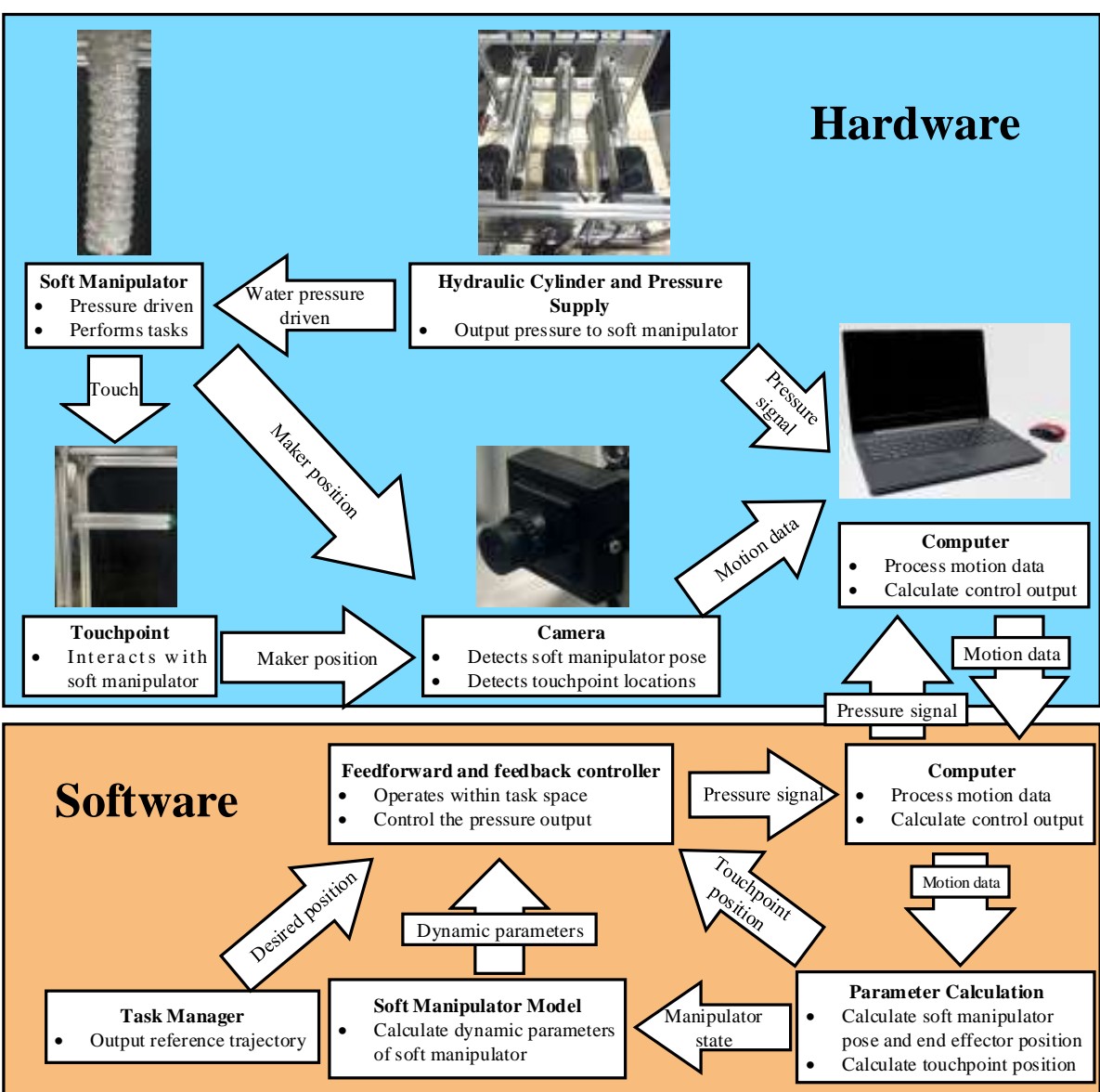

**Figure 10.** Control flow of experimental system.

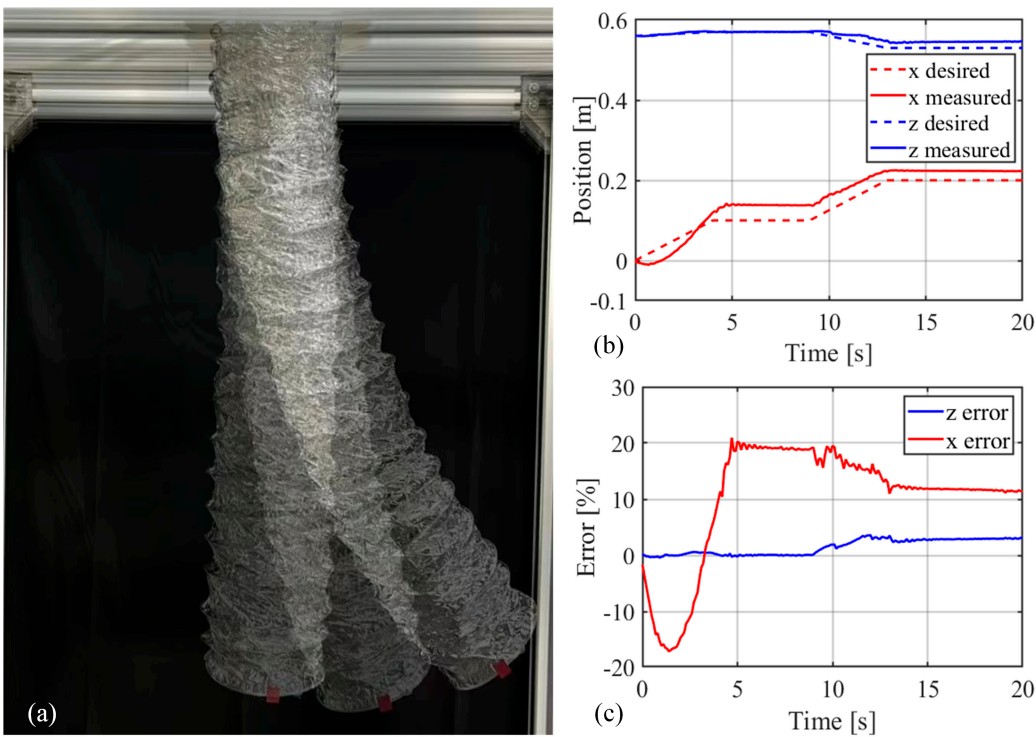

**Figure 11.** One-way trajectory-tracking experiment. (**a**) Motion trend of soft manipulator. (**b**) Comparison between the expected value and the measured value of the end point of the soft manipulator. (**c**) The trajectory error of the end point of the soft manipulator.

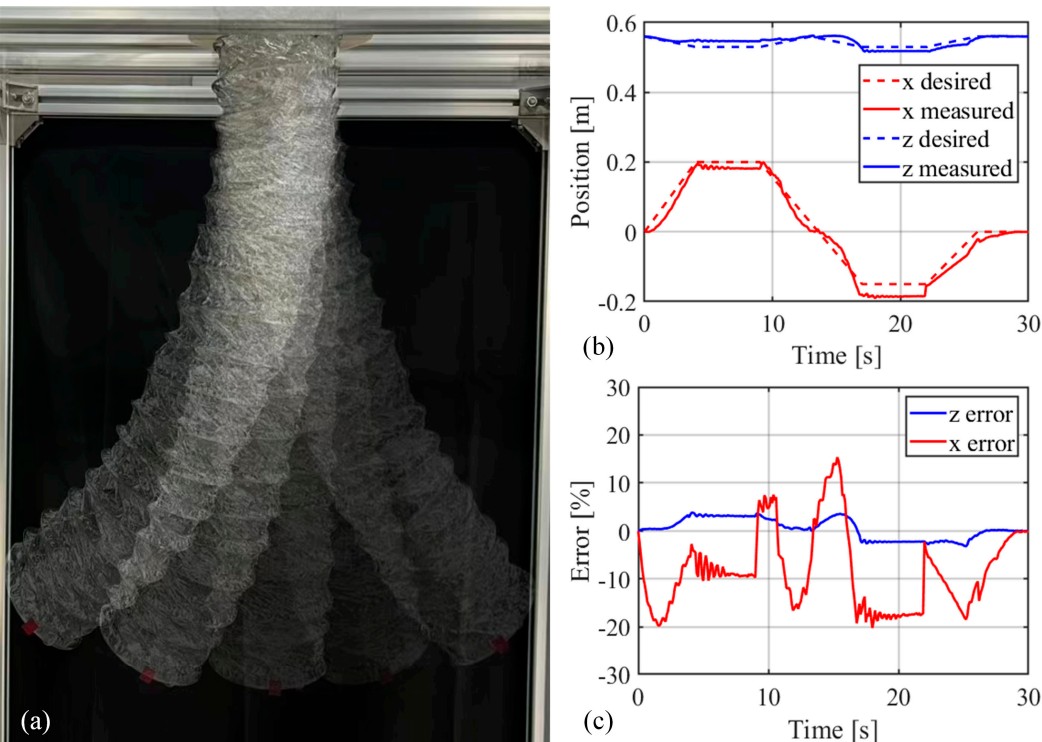

**Figure 12.** Swing track tracking experiment. (**a**) Motion trend of soft manipulator. (**b**) Comparison between the expected value and the measured value of the end point of the soft manipulator. (**c**) The trajectory error of the end point of the soft manipulator.

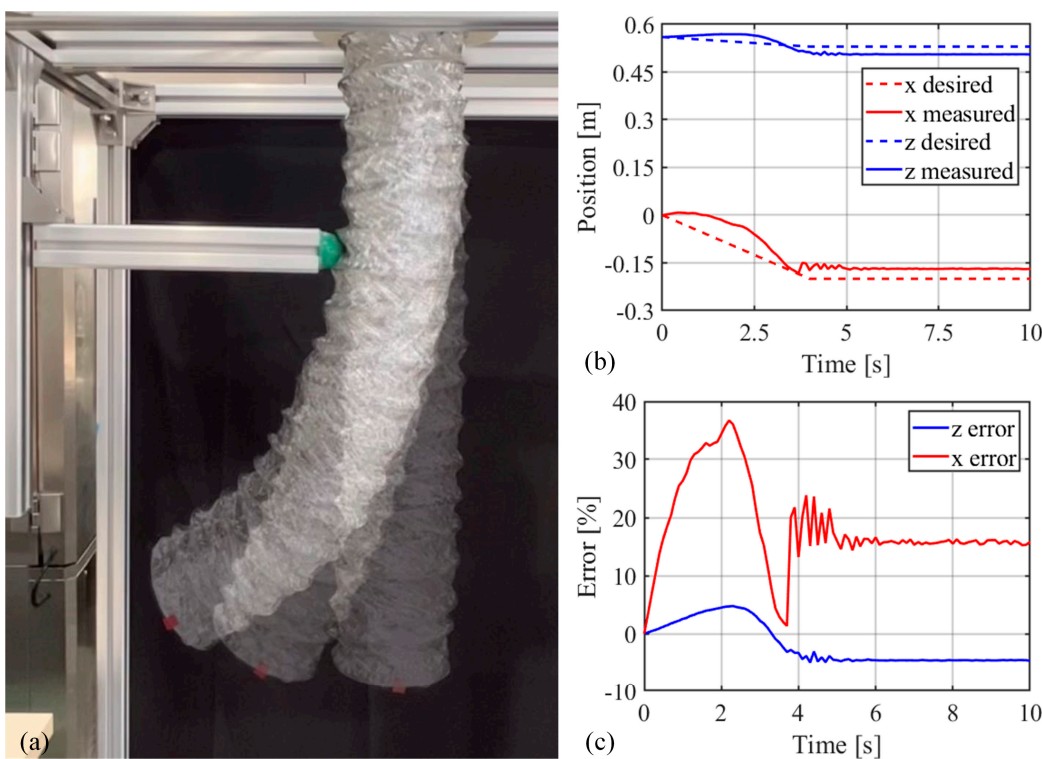

**Figure 13.** The initial static feedforward model is used for collision control experiments. (**a**) Motion trend of soft manipulator. (**b**) Comparison between the expected value and the measured value of the end point of the soft manipulator. (**c**) The trajectory error of the end point of the soft manipulator.

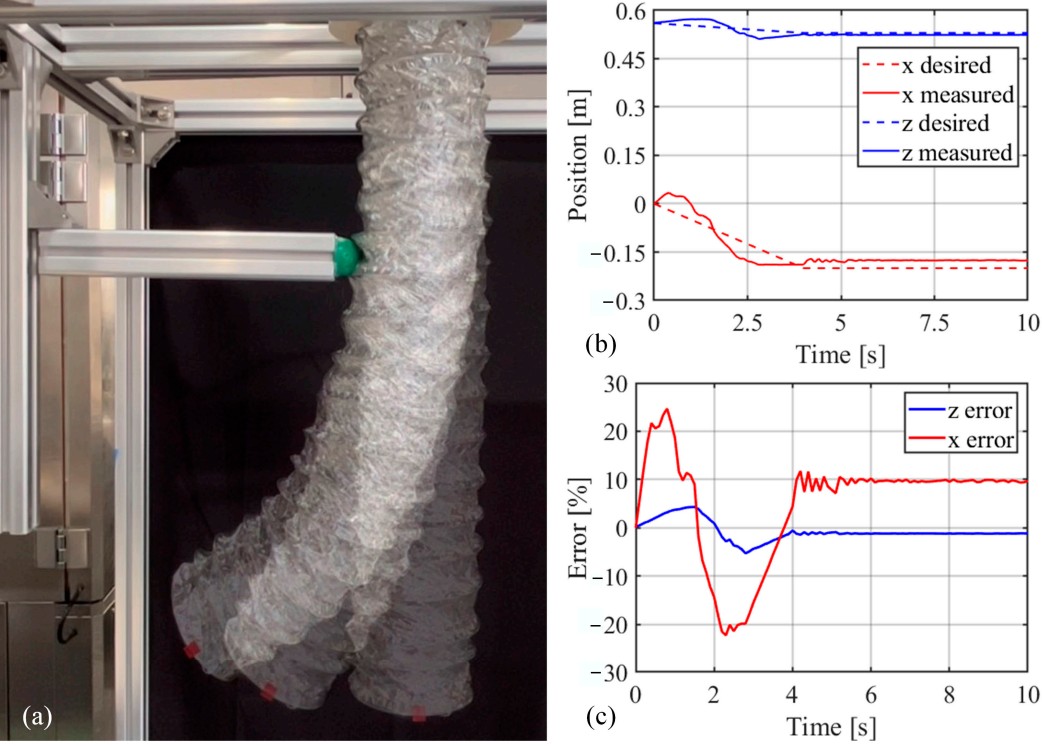

**Figure 14.** The improved static feedforward model is used for collision control experiment. (**a**) Motion trend of soft manipulator. (**b**) Comparison between the expected value and the measured value of the end point of the soft manipulator. (**c**) The trajectory error of the end point of the soft manipulator.

## 5. Conclusions

In this paper, we consider the position control of a soft manipulator. First, we establish a static feedforward model with contact constraints and design a sliding mode controller based on an operational space dynamics model. Then, we combine feedforward compensation and the sliding mode controller for position control under environmental contact, which we simulate and analyze. Finally, we build an experimental platform and carry out a trajectory-tracking experiment with a surrounding environmental contact. The experimental results show that the controller designed in this paper has good performance in real-life application scenarios. The research work carried out in this paper broadens the relevant research in the field of the position control of soft robots. However, these contributions only resolve the position control problem of soft manipulators under environmental contact constraints. In future work, we will study the impedance characteristics of the soft manipulator and the environment.

**Author Contributions:** Conceptualization, methodology, software, Y.C. and Q.S.; validation, Q.S., P.Z. and J.W.; writing—original draft preparation, Q.S.; writing—review and editing, Y.C., Y.G. and J.Z. All authors have read and agreed to the published version of the manuscript.

**Funding:** This research was supported in part by the National Natural Science Foundation of China (NSFC) under Grant 52275053, in part by Fundamental Research Funds for the Central Universities under Grant 3132022352, and in part by the Open Foundation of the State Key Laboratory of Fluid Power and Mechatronic Systems under Grant GZKF-202112.

**Institutional Review Board Statement:** Not applicable.

**Informed Consent Statement:** Not applicable.

**Data Availability Statement:** Not applicable.

**Conflicts of Interest:** The authors declare no conflict of interest.

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
