# Peer review of "Sliding Mode Control with Feedforward Compensation for a Soft Manipulator That Considers Environment Contact Constraints"

_jmse, doi:10.3390/jmse11071438_

Round 1

Reviewer 1 Report

In this manuscript, the authors present a new controller for soft robotic manipulators. The controller is based on a dynamic modeling method that the authors developed and validated previously. However, with this new controller, they implement a feedforward compensation to deal with obstacles (or, more generally, perturbations from the environment). The authors describe the model and controller, test it under simulation, and experimentally with a soft robotic platform. The authors show some improvement in reaching a target in both simulations and experiments. While the work seems thorough, I'm not fully convinced that it is providing a significant improvement. 

Major concerns:

In general, the authors need to better motivate the need for this improved controller, and provide comparisons with previous models/controllers. Below, I provide more specific major concerns, line by line:

- Lines 38-39, how is the kinematics modeling "relatively perfect"? Relative to what? Can you be more specific?

- Line 46-48, the establishment of a new dynamic modeling method isn't really motivated by the statements prior to this sentence. If the previous models were "relatively perfect", then why did this new method need to be developed?

- Lines 77-79, the motivation for the new controller is a bit weak. What is missing? What were the previous controllers unable to do? For example, in lines 72-75, the authors briefly describe one of these previous controllers and it seems that it worked. So why create a new one?

- Figure 1 is quite unclear. There are a lot of variables and lines overlaid and some of them cannot be read clearly. It is also difficult to see what some of the variables refer to. Also, why is the segment i rotated by 180 degrees between panels (a) and (b). I don't seem to understand the differences between li1, li2, li3 and ri, ri1.

- For the 'Elastic Force and Torque' subsection (from line 149), shouldn't the authors state that they are assuming the material of the actuator is Hookean? Is this assumption true even for large deformations? Can the authors comment on this?

- From Figure 3, there is a step that requires the user to 'Enter obstacle Location(area2)'. Does this mean that the obstacle locations have to be known a priori? If so, then to me it doesn't really seem to be such a useful controller.

- Also, in Figure 3, how come there is no 'critical state' in it? In lines 185-186, the authors mention three operational states, including noncontact, critical, and contact. Figure 3 shows the noncontact and contact states, but not the critical one. Does the critical state just use the noncontact workflow? If so, then what is the point of differentiating between noncontact and critical states?

- It is a bit unclear to me how Equation 24 is obtained, especially given Equation 19.

- I am confused as to what the difference is between the controller used to obtain the results shown in Figures 6 & 7. Since there is no obstacle, then the controller should just be the contact-less feed-forward controller, as specified in Figure 3. So what is the difference? Why does one account for gravity and the other not?

- Lines 425-427, it is difficult for me to agree that there is 'good' trajectory tracking when there are errors as large as 66.7%. Can the authors compare with other controllers in order to justify this 'good' performance?

- In general, I think it is better to plot the error in Figures 6, 7, 8, 9, 11, 12, 13, 14 in terms of % and not length (meters). In other words, Equation 27 should be normalized. Presenting error in length is a bit misleading. Also, in Figures 13 and 14, how are there errors in x and z when Equation 27 includes the three dimensions (x, y, z)?

- When comparing Figures 13 and 14, the errors in x don't seem to improve with the new controller. Can the authors comment on why they think that is?

Minor issues:

- Lines 30-31, I don't understand what is meant by 'bionic principle' and why it only includes soft worms, robotic fish, and soft manipulators. There are a lot more examples of soft robots beyond these three categories.

- Line 61, the 'Yet' seems out of place, maybe a typo.

- For Equation 1, it wasn't immediately apparent to me that the variable p represents the pressure. The variables are introduced in Equation 1, but only explicitly explained in line 146.

- In Equation 10, it is a bit confusing to me that sometimes the segments are represented by numbers (ith) and other times by letters (a, b, c, etc). Or am I misreading the equations?

- Line 255, the 'let' seems out of place, maybe a typo.

- The paragraph starting from Line 312 is a bit unclear to me. It is stated that the soft manipulator is divided into two constant curvature arcs, but then in the next sentence it is stated that it is divided into four constant curvature arcs. Which is it?

- In Equation 28, I'm not really sure what the difference between kx, ky and r is. I thought the r variable defined in Figure 1 was the radius of curvature.

- Starting from Figure 6, all the plots are really blurry and it is difficult to make out text. Also, the pictures in Figure 10.

There are some minor issues, but, overall, the English is okay.

Reviewer 2 Report

In the referring paper, a sliding mode controller is designed to control a particular soft manipulator. Together with the developed static feedforward model involving the contact constraints effect, this particular controller can significantly improve the control accuracy and reduce the trajectory error as elaborately illustrated in the simulation and experiment tests. The developed static feedforward model and sliding mode controller provide reference to similar studies in the field. The paper structure is arranged in a tidy shape and the wording flow also goes very well. The readability of the paper is good. I think it can be accepted for publication in the JMSE. The followings are some suggestions to the authors for further improving the paper in the final revision.

1)     In the introduction, the challenges of considering the environmental contact effect in controller design for soft manipulators should be explicitly discussed (mentioned on lines 78-79).

2)     In Figure 1, the deflection angle phi_i seems problematic. The angle between a vector and a plane is usually measured using a plane normal vector. However, in Figure 1, phi_i shows to be the angle between the x-axis and the lower edge of plane B.

3)     In Equation 3, the right-hand term seems problematic. It is not clear how the transformation matrix converts [0,0,0,1]’ to [x,y,z,1].

4)     In Figure 2, the driving pressure is missing.

5)     Also, it is not clear why the driving pressure always acts along the central axis of the soft unit (on line 141). As described in the paper, the pressure should be uniformly applied on the inner surface of the soft unit. So, the pressure direction should be normal to the inner surface. 

6)     In Equation 6, the driving force is also a vector. Why is it denoted as a scalar? 

7)     On line 424, why does the small displacement fluctuation produce a large relative error? Also, on line 461, Why are individual errors can be ignored?

8)      There are a few typos/writing errors. For example, on line 49, “s” should be appended to “model.” The sentence on line 90, “the simple kinematic feedforward is limited by the kinematic model” is not clear. On line 140, “under” may be replaced by “in.” On line 155, “s” should be appended to “represent.” On line 379, “that is” may be added before “consistent.” On line 421, “when driving to cause errors” is confusing. On line 530, the letter "D" is missing in the first word of the paper title. 

9)     The derivation of X=f_{xg}(g) from equations (2) and (3) should be discussed.

10)   The engineering significance of the developed control methods should be discussed in either introduction or conclusion. Are they only limited to the particular soft manipulator? 

The English writing is pretty fluent and easy to follow. There are only a few writing errors in the sentences or phrases, as listed in the above comments to the authors. 

Reviewer 3 Report

The robot description (that only appeared on page 14) must appear just after the state of the art and before the control explanation. I understood that authors probably want to present their control not based on the specific robot but as a general approach. However, pictures 1,2 and 4 show their model and Table 1 shows the parameters of the real robot. Moreover, Table 1 is not cited in the text, and I would also like to add that it would be better to add a column with the description of each parameter; some of them are pretty obvious, but others are not, so it will be helpful, especially for geometrical parameters to remember what they represent. (line 389 authors mention that the robot is made of "silica gel", it could be better to add some material characteristics)

Can this control be applied to a similar robot with a different actuation? Or is it just for pneumatic ones? It could be good to specify. (I think so because of the previous paper, but it could be good to add in any case).

In lines 192-193, the authors say: "We input the desired end point position of the soft manipulator and the position of the obstacle." How is it possible to know the obstacle's position in a real environment?

In lines 201-202 authors say: "After judging the operational state of the soft manipulator if there is no contact between the soft manipulator and the obstacle, the static feedforward model is used to control the motion of the soft manipulator." From the intersection between a1 and a2, we know whether there will be a contact, right?

However, in the case of contact, we also assume that it is impossible to actuate the manipulator to avoid contact. Does the control calculate any possible trajectory? Moreover, can the manipulator not reach a position due to the obstacle? What will happen in this case?

The authors say they assume there is only one contact point. Moreover, if there is more than one, what happens? Will it not be better to calculate another trajectory to avoid the obstacle?

In line, 239 authors say: "In practical applications, the environment will interfere with the soft manipulator." However, previously, they said that they input the obstacle position, so I am still determining how the system works. Do we know of the obstacle position? Moreover, if you want to control the robot in a real environment without knowing it, why did the authors previously say they introduced it?

Finally, it needs to be clarified how a robot knows it is touching something. Moreover, how does it know that the trajectory diverges due to an obstacle and not another reason?

It could also be good to improve the quality of some pictures, especially figures 6, 7 and 11, 12, 13, 14 b and c.

Round 2

Reviewer 1 Report

The author's have addressed my concerns. The manuscript is publishable as is.

Reviewer 3 Report

The authors address all the questions well, and I consider the paper ready for publication. I have no more comments to add.